# Evolution of Pharmacological Treatments and Associated Costs for Multiple Myeloma in the Public Healthcare System of Catalonia: A Retrospective Observational Study

**DOI:** 10.3390/cancers15225338

**Published:** 2023-11-09

**Authors:** Gemma Garrido-Alejos, Guillem Saborit-Canals, Laura Guarga, Thais de Pando, Miriam Umbria, Albert Oriol, Anna Feliu, Caridad Pontes, Antonio Vallano

**Affiliations:** 1Department of Pharmacology, Therapeutics and Toxicology, Autonomous University of Barcelona, 08193 Barcelona, Spain; ggarrido@gencat.cat (G.G.-A.); cpontes@catsalut.cat (C.P.); 2Medicines Strategy and Coordination Unit, Catalan Health Institute, 08007 Barcelona, Spain; 3Medicines Department, Catalan Health Service (CatSalut), 08007 Barcelona, Spain; gsaboritcanals@gmail.com (G.S.-C.); laura.guarga@catsalut.cat (L.G.); thais.pando@catsalut.cat (T.d.P.); mumbria@catsalut.cat (M.U.); 4Digitalization for the Sustainability of the Healthcare System (DS3), Bellvitge Biomedical Research Institute (IDIBELL), 08908 Hospitalet de Llobregat, Spain; 5Catalan Institute of Oncology, Hospital German Trias i Pujol, 08916 Badalona, Spain; aoriol@iconcologia.net; 6Pharmacy Department, Hospital de la Santa Creu i Sant Pau, 08025 Barcelona, Spain; afeliu@santpau.cat; 7Healthcare Management of Hospitals, Catalan Health Institute, 08007 Barcelona, Spain

**Keywords:** multiple myeloma, hematological neoplasms, pharmacological treatments, health expenditures, treatment patterns

## Abstract

**Simple Summary:**

In this study, we examined the prescription patterns and pharmaceutical costs for treatments of multiple myeloma (MM) in Catalonia’s public healthcare system over the past decade. MM is a complex and costly disease for which substantial pharmacological innovation has recently been introduced, and we aimed to understand how treatment choices and expenses have evolved. We analyzed data from 4556 MM patients and found that the number of treated patients increased annually. Drugs like lenalidomide and daratumumab showed significant utilization rises, impacting overall costs. Our findings shed light on the economic burden of MM treatment, emphasizing the importance of monitoring and optimizing healthcare resources allocation for better healthcare decision making.

**Abstract:**

(1) Background: Our understanding of and treatment for multiple myeloma (MM) has advanced significantly, and new pharmacological treatments have promising benefits but high price tags. This study analyzes prescription patterns and pharmaceutical expenditure for MM treatments in Catalonia’s public healthcare system over eight years. (2) Methods: A retrospective observational study examined MM treatment data from 2015 to 2022 in Catalonia, using healthcare registries from the Catalan Health Service to collect information on patients, medicines used, and treatment costs. (3) Results: A total of 4556 MM patients received treatment, with a rising trend in the number of treated patients each year from 902 in 2015 to 1899 in 2022. The mean age was 68.9 years, and patients were almost evenly distributed by gender (51.5% male). Most patients were treated with bortezomib (3338 patients), lenalidomide (2952), and/or daratumumab (1093). Most drugs showed increased utilization annually, most significantly for lenalidomide and daratumumab. The total pharmacological treatment cost throughout the entire study period was EUR 321,811,249, with lenalidomide leading with the highest total cost (EUR 157,236,784), and daratumumab exhibiting the highest increase in annual expenditure. (5) Conclusions: The study reveals a progressive increase in the number of MM patients treated and rising pharmaceutical costs. Lenalidomide and daratumumab incurred the highest costs. The findings highlight MM treatment’s economic impact and the need to monitor prescription patterns and expenditures to optimize healthcare resources and decision making. Understanding these trends can guide resource allocation effectively.

## 1. Introduction

Multiple myeloma (MM) is a hematologic neoplasm characterized by the clonal proliferation of malignant plasma cells in bone marrow, monoclonal immunoglobulin production, and associated organ disorders [1,2]. It is the second most frequent hematological malignancy and accounts for 1–1.8% of all cancers [3]. In Europe, the estimated age-standardized incidence and mortality rates for 2020 were 6.8/100,000 and 4.4/100,000, respectively [4]. However, MM is a heterogeneous disease with survival ranging from a few months to over a decade [5]. In Spain, the estimated total prevalence in 2020 was 16,307 patients, and the number of new cases estimated for 2023 was 3082 [6].

The recommended treatment for newly diagnosed MM patients under 70 years old without comorbidities is an induction regimen aimed at achieving remission, followed by high-dose therapy (HDT) with autologous stem cell transplantation (ASCT) and maintenance pharmacological treatment [3]. For elderly patients or those not eligible for HDT and ASCT, as well as for patients with relapsed or refractory disease, different drug regimens are available. Most MM patients will experience disease relapse, and then may undergo multiple lines of therapy [7].

Over the past 10 years, several novel agents have been incorporated into MM therapeutic strategies, leading to significant improvements in patients’ survival [3,8,9]. Currently, various drug classes are available for the treatment of MM, including cytostatic/cytotoxic agents, corticosteroids, proteasome inhibitors (PIs), immunomodulatory agents (IMiDs), or monoclonal antibodies (mAbs) targeting different surface antigens. While some drugs are used in monotherapy, agents from different classes are often combined in doublet, triplet, or even quadruplet regimens. Furthermore, the costs of MM treatment have also increased in recent years [10,11]. The improvement in survival, the introduction of high-cost novel agents, the earlier use of expensive drugs initially authorized only for relapsed or refractory disease, and the recommendation of triplet or quadruplet regimens as first-line therapy may have influenced MM expenditure. Moreover, it is anticipated that chimeric antigen receptor (CAR)-positive T cell therapies and bispecific antibodies will be added to the therapeutic options in the near future, leading to further increases in treatment costs. In Spain, most drug regimens approved in Europe for MM treatment are reimbursed by the Spanish National Health Service (NHS) [12].

In general, the impact of new therapies needs to be evaluated, both regarding changes in patient management and associated clinical outcomes, as well as regarding the cost of treating these hematologic neoplasms, with a general healthcare system perspective. Real-world studies are emerging as valuable tools for collecting data from daily clinical practice, assessing the impact of new interventions, providing support in treatment decision making and guiding the implementation of healthcare measures [13,14,15,16]. Such a perspective is especially relevant in healthcare systems with universal public coverage such as the Spanish NHS.

An analysis of MM pharmacological treatments could help characterize their use, costs, and the impact on the public healthcare budget. While some studies have described MM treatment patterns and costs [17,18,19,20,21,22], few of them included data after 2019 [22], and none were conducted in our setting. The detailed clinical and administrative data available in the region of Catalonia, providing universal coverage to 16% of the Spanish population, represents an opportunity to analyze valuable information [23]. Thus, this study aims to analyze the trends of MM drug use and expenditure in Catalonia over the recent years.

## 2. Materials and Methods

This retrospective observational study followed STROBE criteria [24] and used data from different Catalan healthcare registries to assess treatment patterns and costs from 2015 to 2022.

### 2.1. Data Source

The Catalan Health Service (CatSalut) centrally manages all healthcare registries in Catalonia, collecting clinical practice data on 63 hospitals and more than 1600 primary healthcare centers that provide public universal care to 7.9 million people [25,26]. For this study, we combined data from the following healthcare registries [23,27]:The Datamart Billing Service, which collects all of the specific billing data from healthcare providers, including hospital outpatient drugs dispensed through hospital pharmacies;The registry of patients and treatments of hospital outpatient medicines (RPT-MHDA), which contains clinical information on drugs prescribed from different therapeutic areas and dispensed through hospital pharmacies;The Datamart Electronic Prescription, which collects information of all prescribed drugs with electronic prescriptions and dispensed through community pharmacies;The Central Registry of Insured Persons, which contains basic demographic data on the population insured by CatSalut.

### 2.2. Study Population

The inclusion criteria aimed to encompass all patients with a unique personal identification code diagnosed with MM who were administered specific standard treatment regimens for the condition, as recorded in the Datamart Billing Service. Additionally, medicines prescribed as part of these treatment regimens, listed in the Datamart Electronic Prescription, were also included. The RPT-MHDA registries were manually reviewed to exclude patients with indications other than MM.

The utilization of personal identification codes facilitated the integration of patient data from multiple treatment centers, thereby preventing the duplication of patients treated across different hospitals.

### 2.3. Outcomes

A descriptive analysis was conducted on patient demographic characteristics, including age at initial treatment registration and sex, number of treated patients, and pharmaceutical expenditure. The medicines encompassed in the analysis consisted of hospital outpatient drugs reimbursed for MM treatment (bortezomib, carfilzomib, daratumumab, pegylated liposomal doxorubicin, isatuximab, lenalidomide, pomalidomide, and thalidomide) along with drugs dispensed at community pharmacies that are administered concomitantly (dexamethasone, prednisone, and oral melphalan). Medications pending a reimbursement decision or excluded from reimbursement (for example, CAR-T cell therapies or ixazomib, respectively); immunostimulants; medicines for managing MM-related complications; those administered during hospitalization; and clinical trials drugs were not considered in the study.

The analysis encompassed both the number of patients and pharmaceutical expenditure for the entire study period (2015–2022), broken down annually, and was conducted for all included medicines collectively and individually for each medicine. The annual count of patients accounted for both prevalent and incident cases. The cost per patient was determined for all included patients, without imposing any restriction on the duration of follow-up, and this calculation was performed annually. Subsequently, subgroup analyses were carried out based on the number of medicines prescribed (1, 2, 3, or ≥4) for both the annual patient count and the annual cost per patient.

For pharmaceutical expenditure, we considered the data collected in the Datamart Billing Service and the Datamart Electronic Prescription, which use the NHS reimbursed price. Hospital outpatient drugs are fully paid by CatSalut, and drugs dispensed through community pharmacies are included in a pharmaceutical copayment system. For this study, only the costs paid by CatSalut were considered.

### 2.4. Statistical Analysis

Descriptive statistics were conducted to summarize the data. Frequencies or proportions were calculated for categorical variables, whereas for continuous variables, means and standard deviations (SDs) or median and interquartile ranges (IQRs) were computed. The statistical analyses were performed using IBM SPSS Statistics for Windows, Version 26.0. Armonk, NY, USA: IBM Corp and Excel for Windows, version 2016. Redmond, WA, USA: Microsoft Corporation.

### 2.5. Ethics

This study obtained approval from the Research Ethics Committee (CRE) with medicines (CREm) at Bellvitge University Hospital before its commencement (ref. EOM017/22). All ethical considerations, encompassing patient privacy and data confidentiality, were thoroughly addressed and adhered to throughout the study.

## 3. Results

A total of 4556 patients were included in the study with a mean (SD) age of 68.9 (12.3) years at their first treatment registration, with 2237 (51.5%) men and 2209 (48.5%) women. The annual number of incident patients remained stable throughout the entire study period, with a mean (SD) of 534 (57) patients (Figure 1). The annual number of prevalent patients showed a progressive increase, from 908 patients in 2015 to 1906 patients in 2022. In 2015, patients treated for MM accounted for 3.5% of all patients with cancer receiving oncology medications, while in 2022, they comprised 4.6% of all treated oncology patients (Table 1).

Out of the 4556 patients treated throughout the entire study period, the distribution of drug treatments was as follows (Table 2): corticosteroids (dexamethasone and prednisone) were used by 3649 patients (80.1%); bortezomib by 3338 patients (73.3%); lenalidomide by 2952 patients (64.8%); daratumumab by 1093 patients (24.0%); melphalan by 977 patients (21.4%); thalidomide by 852 patients (18.7%); carfilzomib by 527 patients (11.6%); pomalidomide by 426 patients (9.4%); doxorubicin by 87 patients (1.9%); and isatuximab by 34 patients (0.7%).

The analysis of the entire study period revealed an annual increase in the number of patients treated with all drugs, except for melphalan, thalidomide, and doxorubicin, which showed a decrease (Figure 2). Notably, the number of patients treated with lenalidomide steadily and progressively increased each year, from 340 patients in 2015 to 1256 patients in 2022. Particularly noteworthy is the rise in the number of patients treated with lenalidomide as monotherapy for maintenance therapy following ASCT. In 2018, this subgroup represented 14% of the total lenalidomide-treated patients (94 patients), reaching 33% in 2022 (410 patients). Furthermore, there was a significant rise in the annual number of patients treated with daratumumab, from 73 patients in 2017 to 601 patients in 2022.

The total cost of pharmacological treatment during the study period amounted to EUR 322.64 M (Table 2). Lenalidomide accounted for EUR 157.2 M (48.73% of the total cost); daratumumab for EUR 65.13 M (20.19%); bortezomib for EUR 56.55 M (17.53%); pomalidomide for EUR 22.08 M (6.84%); carfilzomib for EUR 16.54 M (5.13%); thalidomide for EUR 3.88 M (1.20%); isatuximab for EUR 392.0 K (0.12%); corticosteroids for EUR 374 K (0.12%); melphalan for EUR 347 K (0.11%); and doxorubicin for EUR 104 K (0.03%).

The annual pharmaceutical expenditure has progressively increased from EUR 19.32 M in 2015 to EUR 63.14 M in 2022 (Figure 3). In 2015, pharmaceutical treatment expenses for MM patients comprised 9.8% of the total expenditure for cancer patients undergoing oncology medication treatment (Table 1). By 2022, the pharmaceutical treatment expenses for MM patients had risen to 14.4% of the total expenditure on oncology medications for 4.6% of all treated cancer patients. The lenalidomide cost increased from EUR 9.54 M in 2015 to EUR 30.65 M in 2021, and then decreased to EUR 22.47 M in 2022 (Figure 3). It is noteworthy that the cost of lenalidomide maintenance therapy following ASCT accounted for 10% (EUR 1.74 M) of the total lenalidomide cost in 2018 and increased to 39% (EUR 12.02 M) in 2021, experiencing a slight decrease to 37% (EUR 8.41 M) in 2022. In contrast, bortezomib expenditure decreased from EUR 7.78 M in 2015 to EUR 6.52 M in 2022, and daratumumab expenditure increased progressively from EUR 2.07 M in 2017 to EUR 26.54 M in 2022.

The median (IQR) and mean (SD) total treatment costs per patient (IQR) were EUR 39,603 (EUR 14,357–EUR 102,208) and EUR 70,816 (EUR 79,100), respectively (Table 3). The median annual cost per patient gradually increased from EUR 16,887 (EUR 6587–EUR 30,479) in 2015 to EUR 33,547 (EUR 14,236–EUR 47,100) in 2021, and then decreased to EUR 25,896 (EUR 13,511–EUR 47,431) in 2022.

The annual number of treated patients and the annual cost per patient were analyzed using the number of medicines. The subgroup of patients treated with two medicines was the largest every year, followed by monotherapy from 2015 to 2017 and three medicines from 2018 to 2022 (Table 4). The growth in the number of treated patients was bigger for subgroups with higher numbers of drugs (47% for monotherapy, 77% for two medicines, 199% for three medicines, and 681% for four or more medicines). Finally, the subgroup receiving four or more medicines had the highest median annual cost per patient (Table 5).

## 4. Discussion

The primary objective of this study was to analyze the temporal evolution of prescription patterns and the pharmaceutical expenditure of medicines used in the treatment of MM within the public healthcare system of Catalonia over the past eight years. The mean age of the treated patients was 68.9 years, and there was a slight male predominance, consistent with previous studies [17,19,22,28]. The findings shed light on the increased prevalence of MM patients receiving treatment, the increased utilization of combinations and new drugs, and the increase in associated costs. Furthermore, an increase in the proportion of the pharmaceutical spending dedicated to treat MM patients, in relation to the overall pharmaceutical spending on cancer, has shown a steady increase over the years. This trend suggests a growing burden of MM on the healthcare system, which is consistent, at least in part, with the increasing survival rates associated with the disease [6], as suggested by the sustained increase in prevalent patients along the period.

The number of prevalent treated MM patients and pharmaceutical expenditure in CatSalut increased annually throughout the entire study period. However, these growths were not proportional, as the number of patients increased by 110%, while expenditure increased by 227%. These trends have also been observed across Europe for cancer overall [29]. It is worth noting the huge increase since 2015 in the number of patients treated with three (199% increase) and four or more medicines (681%).

The analysis of prescription patterns showed that the most commonly used drugs for MM treatment were corticosteroids, bortezomib, lenalidomide, and daratumumab. These findings are consistent with the current standard of care for MM, which involves the use of corticosteroids, PI, IMiD, and anti-CD38 mAb [3,30]. Notably, the number of MM patients treated with corticosteroids, lenalidomide, or daratumumab sustainedly increased annually throughout the study period, in contrast with the rest of the medicines, which experienced fluctuations. This may reflect the adoption of these drugs in clinical practice due to their proven efficacy and the availability of funding within the healthcare system.

Lenalidomide, daratumumab, and bortezomib are the medicines with the highest expenditure. These results are similar to those observed in previous studies conducted in other European countries [21,22]. In 2015, there were four authorized treatment regimens in Europe that included lenalidomide, while in 2022, there were ten [9]. Moreover, since its approval in 2017, lenalidomide monotherapy is the only MM maintenance treatment after ASCT available in Europe. However, expenditure on lenalidomide does not follow the same trend as the number of treated patients, since it plummeted from EUR 30.65 M in 2021 to EUR 22.47 M in 2022. The main cause of this drop-off is the introduction of generic products of lenalidomide since December 2020 [31]. Notably, the influence and impact of patients undergoing lenalidomide maintenance therapy following ASCT on associated costs should be considered. This subgroup has shown a significant increase in recent years, contributing substantially to the evolving landscape of lenalidomide utilization and its corresponding economic implications. The ideal duration of lenalidomide maintenance therapy is still under debate due to concerns about potential toxicity and costs. It is important to emphasize the necessity for a thorough cost analysis, specifically distinguishing between lenalidomide monotherapy, as maintenance therapy following ASCT, and combination therapies. This differentiation could provide valuable insights into annual maintenance costs, especially considering the introduction of generic alternatives. The decrease in lenalidomide maintenance therapy costs in 2022 compared to 2021 is likely attributed to the introduction of generic lenalidomide. Delving deeper into this trend could offer valuable insights into the evolving economic landscape of lenalidomide treatments post-transplantation.

Currently, the only MM medicines with generics are bortezomib and lenalidomide, which have been commercially available in Spain since September 2018 and December 2020, respectively [31]. The annual expenditure for bortezomib decreased in 2019 after reaching its peak in 2018, and it remained relatively stable thereafter. However, the generics of bortezomib did not appear to have a significant impact on the annual expenditure of MM. In contrast, the generics of lenalidomide probably contributed to a more discrete increase in annual expenditure from 2021 to 2022 than expected based on the previous year’s trends and could also explain why the median cost per patient in 2022 was the lowest since 2018. It is worth noting that in the year following the release of generics, the number of patients treated with bortezomib (2019, 574 patients) was half that of the number of patients treated with lenalidomide (2021, 1130 patients), which could help explain these differences. The impact of generics on cost and expenditure can vary depending on several factors, such as market dynamics, pricing, and availability [32,33,34].

On the other hand, both the number of patients and expenditure on daratumumab rose from 2017 to 2022 and significantly increased in the last two years. Daratumumab was authorized in 2016 as monotherapy for the treatment of relapsed and refractory MM, and since then, it has been approved in combination with other drugs as the first or second line of therapy [9]. By the end of the study period, there were eight MM treatment regimens available that included daratumumab, with three of them indicated for newly diagnosed patients and recommended as the first option in clinical guidelines [3,30]. Consistent with the fact that it was a new launch in the period, daratumumab had the highest increase in annual expenditure throughout the entire study period, both in absolute terms (EUR 24.47 M) and in relative terms (1184%). Yet, although the general rule for pricing is that when more indications are reimbursed the unit prize should reduce, and despite the fact that some pricing agreements were granted in the period, daratumumab was the drug with the highest annual expenditure at the end of the study (EUR 26.54 M).

Future generics and biosimilars of MM medicines could have a significant effect on treatment costs. The orphan market exclusivity for daratumumab’s treatment of MM will expire in May 2026 [35], and it is expected that its biosimilars will make a significant difference in expenditure if daratumumab combinations are still recommended for newly diagnosed patients. Overall, biosimilars offer several economic benefits that can help to reduce healthcare costs, improve patient access to treatments, and free up resources for other healthcare needs [36,37].

However, there are several upcoming MM treatments with a potentially high budgetary impact. The gene therapies idecabtagene vicleucel (ide-cel) and ciltacabtagene autoleucel (cilta-cel) are indicated in Europe for the treatment of adult patients with relapsed and refractory MM who have received at least three prior therapies, including IMiD, PI, and anti-CD38 mAb and have demonstrated disease progression on the last therapy [9]. Both ide-cel and cilta-cel showed clinically meaningful response rates in single-arm trials [38,39] and demonstrated improvements in outcomes compared to currently available therapies in triple-class-exposed relapsed and refractory MM patients, as derived from adjusted indirect comparisons [40,41]. These gene therapies are still awaiting reimbursement decisions in Spain [31], and the expectation is that their costs will be extremely elevated, as ide-cel’s cost in France exceeds EUR 350,000 [42], with financing conditions including payments based on outcomes, similar to other gene therapies [43]. Furthermore, both ide-cel and cilta-cel have demonstrated efficacy in earlier treatment lines [44,45], thus potentially increasing the number of patients who may be candidates for these treatments in the near future. Other MM medicines with a potentially significant impact on MM treatment in the short–medium term include bispecific antibodies teclistamab, talquetamab, and elranatamab. Teclistamab and talquetamab are also awaiting a reimbursement decision in Spain and are expected to influence the current standard of care [46], while elranatamab received a positive opinion from the Committee for Medicinal Products for Human Use on 12 October 2023 and is awaiting for European Comission marketing authorization. 

The percentage of patients annually treated with two or more medicines increased from 63% in 2015 to 74% in 2022. However, a significant number of patients, around one in four in the year 2022, had only received a single medication for the treatment of MM. Lenalidomide as monotherapy is indicated for maintenance treatment after ASCT, and in several MM regimens, one of the drugs is administered alone until disease progression after the first cycles, such as daratumumab when combined with bortezomib, thalidomide, and dexamethasone for the treatment of patients with newly diagnosed MM.

There is a scarcity in the literature of similar analyses including data from the last three years, during which, several regimens with meaningful clinical and economic impacts have been added to the MM therapeutic armamentarium. However, available studies describing MM treatment costs in earlier periods also showed progressive yearly increases [17,20], and multiple studies assessing MM healthcare resource utilization found that medicines were a key cost driver in any line of treatment [18,19,21,47,48,49,50]. The mean cost per patient in a healthcare resource utilization study conducted in France was EUR 61,500 [22], almost EUR 10,000 lower than the figure reported here (EUR 70,816). However, the patient characteristics and analyzed periods are not comparable, as the French study included patients newly diagnosed with MM from 2013 to 2018 and followed them up for a median of 22 months, whereas our study included all MM patients treated from 2015 to 2022. A study that assessed the burden and cost of MM in Portugal in 2018 estimated average pharmacological expenditure of EUR 25,348 per patient/year [21], aligning with the cost per patient in 2018 reported here (mean: EUR 28,125; median: EUR 23,270). Finally, a report published in 2018 estimated the total annual cost of MM in Spain to be around EUR 940 M, with approximately EUR 250 M attributed to pharmacological treatment [51]. This estimation is consistent with the results of our study, considering that the Catalan population represents 16% of the Spanish population [52].

The increasing pharmaceutical costs associated with MM treatment observed in this study can be explained by the availability of new medications and advancements in therapeutic options. The rising costs of pharmacological treatments pose challenges to healthcare systems, as they need to balance the access to effective therapies with financial sustainability and equity. Future studies should explore not only the cost impact, but also patient outcomes, such as assessing the cost-effectiveness of different treatment sequences or evaluating the potential benefits of treatment optimization algorithms.

Understanding the temporal evolution of prescription patterns and pharmaceutical expenditure for MM treatments has significant implications for healthcare resource allocation and decision making. Drugs introduced in reduced niches of refractory disease may evolve to become part of treatment combinations and advance from late to earlier phases of disease, posing challenges at the time of first pricing and reimbursement decisions, which thereafter set a high base price that is difficult to re-negotiate, especially since the actual use of new indications and combinations is difficult to contain once new products are in the market.

The findings of this study underscore the economic impact of MM and highlight the need for the ongoing monitoring of prescription patterns and expenditures. Through the analysis of treatment utilization trends and cost data, healthcare authorities can identify opportunities to optimize resource allocation and implement cost-effective measures. This may include interventions to negotiate fair prices and managed access agreements, to promote the rational use of medications, such as developing guidelines to support evidence-based and cost-conscious treatment decisions, supported by pharmacological tariffs.

The present study has several notable strengths and limitations. On one hand, there are several strengths worth highlighting in the study, including a large sample size resulting from the exhaustive information from all patients with MM treated within the public healthcare system of Catalonia, which comprises 7.9 million people with universal healthcare coverage. This substantial sample size provided a robust basis for conducting analyses and extrapolating conclusions. The prolonged time period, as the study spanned an eight-year period, from 2015 to 2022, enabled the analysis of temporal trends in prescription patterns and pharmaceutical expenditures in MM treatment. This long-term perspective allowed us to gain a good understanding of changes and patterns over time. The study benefitted from representative and reliable data sources because the data utilized in this study were derived from comprehensive patient and treatment records within Catalonia’s public healthcare system, ensuring that results were not estimated through models and ensuring the external validity of the study’s findings. Copayments assumed by patients of drugs dispensed on community pharmacies were not included, but these costs are considered negligible in comparison with the total pharmaceutical expenditure.

On the other hand, there are also several limitations that should be noted. The study was conducted with a retrospective design; thus, the data analyzed were limited to the available items in databases and may have been subject to the inherent limitations and biases associated with this type of design, such as the detail of clinical diagnosis or lines of treatment. Additionally, there is a geographic limitation, as the study focused exclusively on the public healthcare system in Catalonia. Therefore, the generalizability of the findings to other regions or healthcare systems may be limited by the differences in standards of care. Further research in different contexts might be necessary to obtain a more comprehensive picture. The study only focused on pharmacological costs, but a healthcare resource utilization study including hospitalization and other costs would have provided a more comprehensive estimation of the economic burden of MM in Catalonia. Due to the characteristics of the available data, prescription patterns were analyzed for individual medicines rather than treatment regimens, and performing subgroup analyses by line of treatment was not possible. Furthermore, the study did not consider all MM pharmacological treatments, such as immunoestimulants, medicines for the management of MM complications, those administered during hospitalization, and drugs used in clinical trials. An analysis of relevant clinical variables, such as treatment response or patient survival, was not carried out. These clinical aspects are crucial for a comprehensive understanding of the impact of MM treatments. Therefore, logically, a cost-effectiveness analysis was also not conducted. Such analyses would have provided additional insights into the efficiency and value of the medications used. Despite these limitations, the study offers valuable insights into the evolution of prescription patterns and pharmaceutical expenditures in MM treatment in Catalonia. These strengths and limitations should be considered when interpreting and utilizing the study’s findings.

## 5. Conclusions

In conclusion, this study demonstrated a progressive increase in the number of prevalent MM patients treated and an upward trend in pharmaceutical costs within the public healthcare system of Catalonia over the past eight years. Lenalidomide and daratumumab were associated with the highest costs among the analyzed drugs. These findings highlight the economic impact of MM treatment and emphasize the importance of monitoring prescription patterns and expenditures to optimize healthcare resources. The results provide valuable insights for healthcare decision-makers and suggest the need for ongoing evaluation.

## Figures and Tables

**Figure 1 cancers-15-05338-f001:**
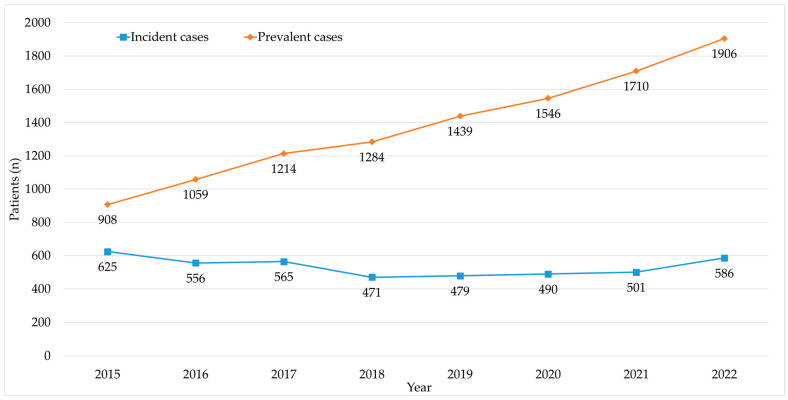
Annual number of treated multiple myeloma (MM) patients (incident and prevalent cases).

**Figure 2 cancers-15-05338-f002:**
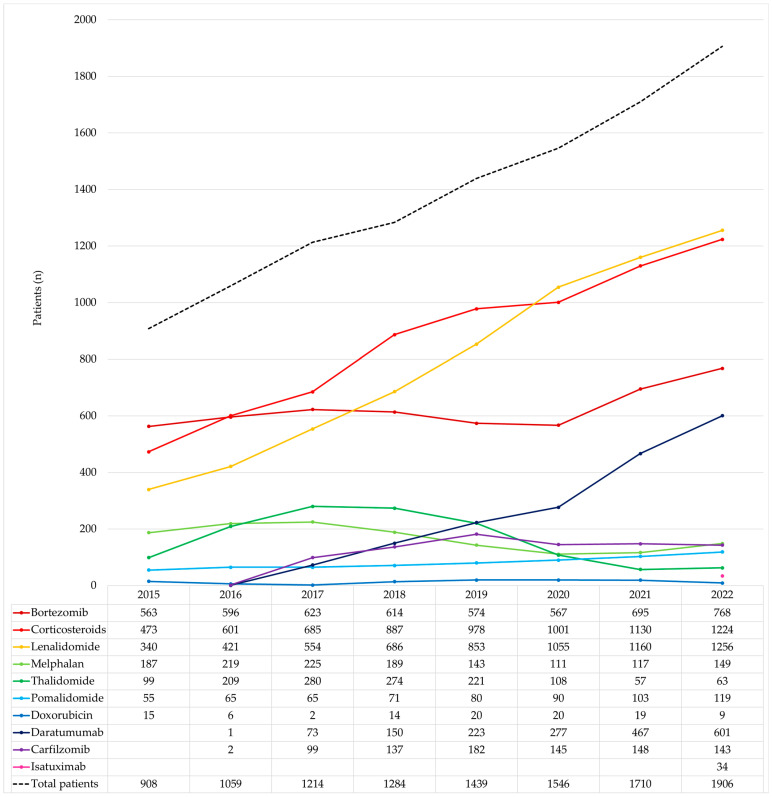
Annual number of MM patients treated with each medicine.

**Figure 3 cancers-15-05338-f003:**
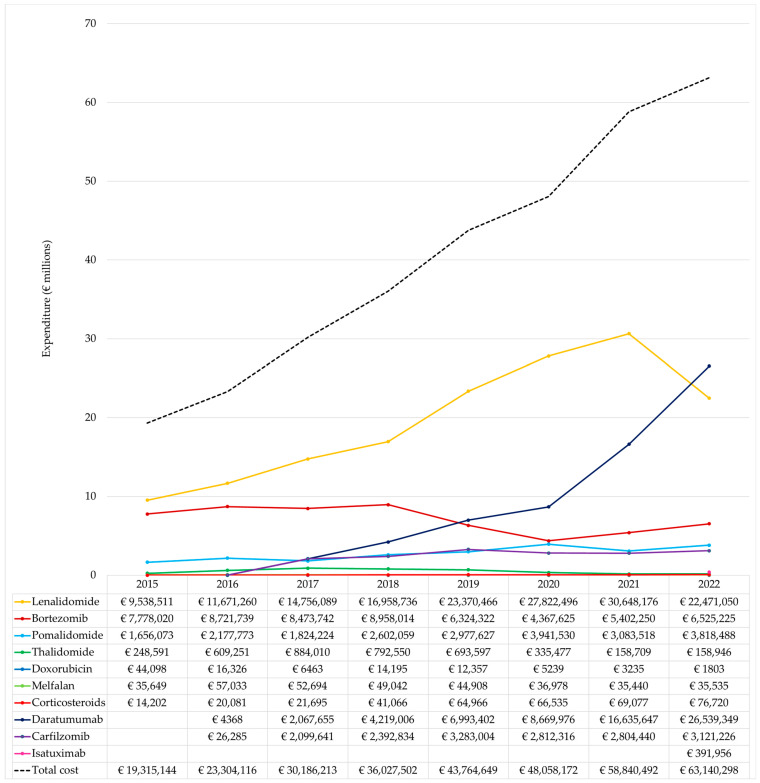
Annual pharmaceutical expenditure of MM for all medicines and by medicine.

**Table 1 cancers-15-05338-t001:** Annual number of multiple myeloma (MM) and cancer patients and pharmaceutical expenditure.

Year	Treated MM Patients	Treated Cancer Patients	MM Patients/ Cancer Patients (%)	MM Pharmaceutical Expenditure	Cancer Pharmaceutical Expenditure	MM Pharmaceutical Expenditure/ Cancer Pharmaceutical Expenditure (%)
2015	908	25,915	3.5%	EUR 19,315,144	EUR 196,170,337	9.8%
2016	1059	29,016	3.6%	EUR 23,304,116	EUR 235,065,906	9.9%
2017	1214	31,350	3.9%	EUR 30,186,213	EUR 273,058,674	11.1%
2018	1284	34,346	3.7%	EUR 36,027,502	EUR 317,120,589	11.4%
2019	1439	36,443	3.9%	EUR 43,764,649	EUR 349,502,597	12.5%
2020	1546	36,795	4.2%	EUR 48,058,172	EUR 387,403,187	12.4%
2021	1710	39,889	4.3%	EUR 58,840,492	EUR 430,940,512	13.7%
2022	1906	41,434	4.6%	EUR 63,140,298	EUR 438,077,749	14.4%

**Table 2 cancers-15-05338-t002:** Cost per medicine and number of MM patients treated with each medicine during the period 2015–2022.

Medicine	Cost	Number of Patients
Lenalidomide	EUR 157,236,784	2952
Daratumumab	EUR 65,129,403	1093
Bortezomib	EUR 56,550,937	3338
Pomalidomide	EUR 22,081,292	426
Carfilzomib	EUR 16,539,746	527
Thalidomide	EUR 3,881,131	852
Isatuximab	EUR 391,956	34
Corticosteroids	EUR 374,342	3649
Melphalan	EUR 347,279	977
Doxorubicin	EUR 103,716	87
Total	EUR 322,636,586	4556

**Table 3 cancers-15-05338-t003:** Total and annual cost per MM patient.

	2015	2016	2017	2018	2019	2020	2021	2022	2015–2022
Patients									
	908	1059	1214	1281	1436	1543	1707	1906	4556
Cost per patient									
Mean	EUR 21,272	EUR 22,006	EUR 24,865	EUR 28,125	EUR 30,477	EUR 31,146	EUR 34,470	EUR 33,127	EUR 70,816
SD	EUR 18,106	EUR 17,400	EUR 20,185	EUR 23,254	EUR 26,157	EUR 23,439	EUR 24,175	EUR 26,227	EUR 79,100
Median	EUR 16,887	EUR 18,374	EUR 20,250	EUR 23,270	EUR 26,216	EUR 28,781	EUR 33,547	EUR 25,897	EUR 39,603
Q1	EUR 6587	EUR 7536	EUR 8799	EUR 10,271	EUR 9615	EUR 11,173	EUR 14,236	EUR 13,512	EUR 14,327
Q3	EUR 30,479	EUR 32,951	EUR 37,112	EUR 40,362	EUR 44,710	EUR 44,989	EUR 47,100	EUR 47,431	EUR 102,208

Q: quartile; SD: standard deviation.

**Table 4 cancers-15-05338-t004:** Annual number of MM patients treated with 1, 2, 3, or ≥4 medicines.

Number of Treated Patients (%)	2015	2016	2017	2018	2019	2020	2021	2022
1 medicine	333 (36.7)	318 (30)	323 (26.6)	226 (17.6)	325 (22.6)	403 (26.1)	417 (24.4)	492 (25.8)
2 medicines	352 (38.8)	450 (42.5)	492 (40.5)	540 (42.1)	583 (40.5)	609 (39.4)	613 (35.8)	622 (32.6)
3 medicines	197 (21.7)	264 (24.9)	318 (26.2)	390 (30.4)	383 (26.6)	405 (26.2)	513 (30)	589 (30.9)
≥4 medicines	26 (2.5)	27 (2.5)	81 (6.7)	128 (10)	148 (10.3)	129 (8.3)	167 (9.8)	203 (10.7)
Total patients	908 (100)	1059 (100)	1214 (100)	1284 (100)	1439 (100)	1546 (100)	1710 (100)	1906 (100)

**Table 5 cancers-15-05338-t005:** Annual cost per MM patient treated with 1, 2, 3, or ≥4 medicines.

Cost Per Patient (EUR)	2015	2016	2017	2018	2019	2020	2021	2022
1 medicine								
Mean	20,154	23,222	22,808	22,742	26,089	29,231	30,756	24,169
SD	20,053	19,878	19,985	19,674	21,235	20,231	17,624	15,910
Median	12,181	17,560	16,690	17,905	27,328	32,920	36,212	21,963
Q1	4334	4829	4184	4595	4381	7997	13,428	10,918
Q3	30,299	39,045	40,419	39,513	44,411	45,411	43,318	36,212
2 medicines								
Mean	19,313	19,571	23,858	25,338	27,243	27,529	29,219	27,150
SD	17,663	16,626	20,832	21,400	23,411	22,424	22,997	23,740
Median	14,563	15,541	18,358	20,598	22,023	23,672	26,777	21,014
Q1	6573	6344	8213	7527	7376	7284	9496	8520
Q3	26,403	28,539	32,799	37,588	43,754	42,320	42,765	37,458
3 medicines								
Mean	24,034	22,724	24,025	30,101	34,901	35,373	36,758	27,150
SD	14,159	14,671	16,942	24,646	31,863	26,977	25,815	23,740
Median	22,719	20,902	20,647	23,199	24,247	27,952	31,804	21,014
Q1	14,161	11,555	12,201	13,995	12,936	15,143	15,903	8520
Q3	31,828	31,084	30,648	39,180	48,715	49,216	51,495	37,458
≥4 medicines								
Mean	38,126	34,424	42,208	42,579	40,266	39,988	55,380	56,236
SD	13,495	16,999	21,005	25,913	25,944	21,563	25,520	28,552
Median	40,585	33,003	41,028	35,951	32,437	38,535	55,884	55,686
Q1	26,828	20,188	23,711	26,446	23,327	22,037	36,233	34,422
Q3	49,516	46,148	53,700	49,466	51,586	55,017	73,241	75,931

Q: quartile; SD: standard deviation.

## Data Availability

The data presented in this study are available on reasonable request from the corresponding author.

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
