# Peer review of "Evolution of Pharmacological Treatments and Associated Costs for Multiple Myeloma in the Public Healthcare System of Catalonia: A Retrospective Observational Study"

_cancers, 2023, doi:10.3390/cancers15225338_

Round 1

Reviewer 1 Report

Comments and Suggestions for Authors

Thanks to the authors for writing this paper. I believe this article would be valuable for policy makers. Please find my comments below:

Line 64: When explaining therapies for MM, it is valuable to mention CAR-T therapies and BsAbs, as they are among the most novel medicines for MM, and some of them are approved by EMA.

A general comment regarding the methodology: Since you only considered the cost of drugs, I assume you may have encountered obstacles in extracting data for hospitalization costs. It would be highly valuable to include hospitalization costs along with drug costs. This addition would significantly enhance the paper, if possible.

In Table 1, the header of column 7 needs modification.

Regarding the figures, it is better to place the titles of the figures after them.

Author Response

Reviewer 1 comments and responses

Comment 1: Line 64: When explaining therapies for MM, it is valuable to mention CAR-T therapies and BsAbs, as they are among the most novel medicines for MM, and some of them are approved by EMA.

Response: The authors would like to clarify that drug classes enumerated on this sentence do not pretend to be an exhaustive list. Moreover, at the moment CAR-T therapies and Bispecific antibodies (BsAbs) are not really available in Spain because they are pending of reimbursement decision.

In response to the reviewer suggestion, the authors have added CAR-T therapies and BsAbs to line 74, which talks about upcoming therapies.

Comment 2. A general comment regarding the methodology: Since you only considered the cost of drugs, I assume you may have encountered obstacles in extracting data for hospitalization costs. It would be highly valuable to include hospitalization costs along with drug costs. This addition would significantly enhance the paper, if possible

Response: The authors acknowledge that analysing hospitalization costs would be highly interesting. However, it is out of the scope of the current study, and they gather the suggestion for a future article.

In response to the reviewer suggestion, the authors have emphasized on the discussion that hospitalization cost where not included in the analysis. 

Comment 3. In Table 1, the header of column 7 needs modification.

Response: The authors would like to thank the reviewer for the correction. The header of column 7 has been modified.

Comment 4: Regarding the figures, it is better to place the titles of the figures after them.

Response: The authors would like to thank the reviewer for the correction. The titles of the figures have been placed after them, according to Cancers journal template.

Reviewer 2 Report

Comments and Suggestions for Authors

The article is of undoubted inerest for practicing hematologists and healthcare specialists. The emphasisi placed by the authours of the article on the cost of drug treatment for patients with MM makes us think about the adequacy of the choice of treatment benefit and the duration of therapy, as well as the correcteness of assessing the effectiveness of therapy. The Disscusion section seemsd somewhat drawn out and, as a result, vague. In general the article makes good impression, and I hope the graphs will be used in the reports of individual hematologists. 

Author Response

Reviewer 2 comments and responses

Comment 1: The Discussion section seemed somewhat drawn out and, as a result, vague.

Response: The authors has shortened and condensed a section of the discussion in response to suggestions from Reviewer 2.

Reviewer 3 Report

Comments and Suggestions for Authors

This is  a large retrospective real-world analysis of the use and cost of myeloma drugs, based on the healthcare registries of a large Spanish region and covering a large period of time (from 2015 to 2022). This period is of special interest since it corresponds to the introduction of novel and expensive drugs and to the development of 3 or 4 drug combinations which yielded a major improvement in survival and therefore in the prevalence of myeloma

Since the patients outcome is not known, it was not possible to perform a cost-effectiveness analysis but this study provides several useful informations

- The annual increase of myeloma prevalence (from 906 in 2015 to 1908 in 2022) mostly due to the improved survival, while the incidence remains stable. As a consequence the proportion of myeloma treated patients within the overall cancer treated patients increased from 3.5 % to 4.6 %

- The increase of drug expenditures during this period (X3.3) and accordingly the increased proportion of myeloma drug expenditures /total cancer drugs expenditures (from 0.8% to 14.4 %)

- The increased number of patients treated with lenalidomide, bortezomib and daratumumab but the reduced annual costs of bortezomib and lenalidomide after the introduction of generics

General comments

1) The analysis of the causes of this increase of both the prevalence and the expenditures is correct . However the impact of long maintenance therapy is not well estimated. In the great majority of cases monotherapy corresponds to lenalidomide maintenance therapy which significantly contributes to the survival improvement . However the optimal duration of this maintenance remains a matter of debate, because of its potential toxicity but also because of its cost. This should be better underlined and for instance, if possible the cost of lenalidomide monotherapy should evaluated separately (compared to lenalidomide as part of combinations). This could provide interesting finding on the annual cost of maintenance after introduction of the generic

2) As a consequence, the part of discussion devoted to monotherapy is not relevant . Even in frail or relapsed patients, combinations are usually prefered (for instance with low-dose corticosteroids). Daratumumab monotherapy has been used for a short period, just after approval and before the introduction of combinations which are much more effective. In end-stage patients monotherapy with alkylating agents (oral melphalan or cyclophosphamide ) can be used but obvioulsy for short periods . This part should be deleted or at least much reduced

3 Ixazomib , the third proteasome inhibitor was not evaluated while it can be prescribed in elderly or relapsed patients because of this relatively low toxicity. Is it because it is not reimbursed in Spain ?

Minor comments

1) Introduction Paragraph 2. Elderly patients are not treated with "numerous regimens" but with 3 different regimens : MVP (mostly in Spain) Rd and VRd. Recently daratumumab was added to MVP and Rd

2) Paragraph 3 Monoclonal antibodies do not target pathways but surface antigens (mostly CD38)

3) Paragraph 3 line 5 redundancy : some drugs are used in monotherapy

4) Paragraph 3 line 12 and discussion CAR T cells are not really gene therapies but rather immunological cell therapies

Author Response

Reviewer 3 comments and responses

Comment 1: The analysis of the causes of this increase of both the prevalence and the expenditures is correct. However the impact of long maintenance therapy is not well estimated. In the great majority of cases monotherapy corresponds to lenalidomide maintenance therapy which significantly contributes to the survival improvement. However the optimal duration of this maintenance remains a matter of debate, because of its potential toxicity but also because of its cost. This should be better underlined and for instance, if possible the cost of lenalidomide monotherapy should evaluated separately (compared to lenalidomide as part of combinations).

This could provide interesting finding on the annual cost of maintenance after introduction of the generic

Response: We recognize the need for a more comprehensive analysis of the impact of long-term maintenance therapy with lenalidomide following autologous stem cell transplantation (ASCT). The suggestion to separately assess the cost of lenalidomide monotherapy after ASCT, as opposed to combinations, has been integrated. We analysed the data on patients undergoing lenalidomide maintenance therapy after ASCT and conducted an evaluation of its costs.

We incorporated commentary on the progressive increase in the percentage of patients receiving lenalidomide maintenance therapy after ASCT, as well as their cost. The data now delineate a clear upward trend in patient numbers, providing insight into the evolving landscape of post-transplantation maintenance strategies. Additionally, we presented the escalating costs linked to lenalidomide maintenance therapy over the years, elucidating its financial impact on the healthcare system.

Comment 2: As a consequence, the part of discussion devoted to monotherapy is not relevant. Even in frail or relapsed patients, combinations are usually preferred (for instance with low-dose corticosteroids). Daratumumab monotherapy has been used for a short period, just after approval and before the introduction of combinations which are much more effective. In end-stage patients monotherapy with alkylating agents (oral melphalan or cyclophosphamide) can be used, but obviously for short periods. This part should be deleted or at least much reduced.

Response: In response to the reviewer suggestion, the part of the discussion that address monotherapy is reduced. However, it has been added the interpretation of lenalidomide treatments analysis performed in response to reviewer first comment. 

Comment 3: Ixazomib, the third proteasome inhibitor, was not evaluated while it can be prescribed in elderly or relapsed patients because of this relatively low toxicity. Is it because it is not reimbursed in Spain?

Response: Ixazomib is excluded of reimbursement in Spain, and therefore excluded from this study (line 132). In response to the reviewer comment, examples of MM medicines pending of reimbursement decision or excluded from reimbursement in Spain have been added to the section about material and methods.

Comment 4: Introduction Paragraph 2. Elderly patients are not treated with "numerous regimens" but with 3 different regimens: MVP (mostly in Spain) Rd and VRd. Recently daratumumab was added to MVP and Rd.

Response: In response to the reviewer comment, “numerous” is replaced by “different” (line 60).

Comment 5: Paragraph 3 Monoclonal antibodies do not target pathways but surface antigens (mostly CD38).

Response: In response to the reviewer comment, “pathways” is replaced by “surface antigens” (line 60).

Comment 6: Paragraph 3 line 12 and discussion CAR T cells are not really gene therapies but rather immunological cell therapies.

Response: The European Medicines Agency considers that CAR-T cells such as idecabtagen vicleucel or ciltacagtagene autoleucel are gene therapies. Consequently, they are named like this in the article.

https://www.ema.europa.eu/en/medicines/human/EPAR/abecma

https://www.ema.europa.eu/en/medicines/human/EPAR/carvykti

Round 2

Reviewer 3 Report

Comments and Suggestions for Authors

The authors  correctly answered my questions